# Can Forum Play Contribute to Counteracting Abuse in Health Care? A Pilot Intervention Study in Sri Lanka

**DOI:** 10.3390/ijerph16091616

**Published:** 2019-05-08

**Authors:** Katarina Swahnberg, Anke Zbikowski, Kumudu Wijewardene, Agneta Josephson, Prembarsha Khadka, Dinesh Jeyakumaran, Udari Mambulage, Jennifer J. Infanti

**Affiliations:** 1Department of Health and Caring Sciences, Faculty of Health and Life Sciences, Linnaeus University, Hus Vita, S-391 82 Kalmar, Sweden; barshaprem02@gmail.com; 2Jönköping’s County Hospital Ryhov, Women’s Clinic, S-55185 Jönköping, Sweden; anke.zbikowski@rjl.se; 3Department of Community Medicine, Faculty of Medical Sciences, University of Sri Jayewardenepura, Gangodawila, Nugegoda 10250, Sri Lanka; kumuduwije@gmail.com (K.W.); din_lk@yahoo.com (D.J.); mambulage@gmail.com (U.M.); 4Dramapedagogbyrån, S-135 43 Tyresö, Sweden; agneta.josephson@gmail.com; 5Department of Public Health and Nursing, Faculty of Medicine and Health Sciences, Norwegian University of Science and Technology, Postboks 8905, N-7491 Trondheim, Norway; jennifer.infanti@ntnu.no

**Keywords:** maternal and reproductive health services, quality of care, dignity and respect, abuse in health care, obstetric violence, intervention science, participatory theatre, Theatre of the Oppressed

## Abstract

Obstetric violence refers to the mistreatment of women in pregnancy and childbirth care by their health providers. It is linked to poor quality of care, lack of trust in health systems, and adverse maternal and neonatal outcomes. Evidence of interventions to reduce and prevent obstetric violence is limited. We developed a training intervention using a participatory theatre technique called Forum Play inspired by the Theatre of the Oppressed for health providers in Sri Lanka. This paper assesses the potential of the training method to increase staff awareness of obstetric violence and promote taking action to reduce or prevent it. We conducted four workshops with 20 physicians and 30 nurses working in three hospitals in Colombo, Sri Lanka. Participants completed a questionnaire before and three-to-four months after the intervention. At follow-up, participants more often reported that they had been involved in situations of obstetric violence, indicating new knowledge of the phenomenon and/or an increase in their ability to conceptualise it. The intervention appears promising for improving the abilities of health care providers to recognise obstetric violence, the first step in counteracting it. The study demonstrates the value of developing further studies to assess the longitudinal impacts of theatre-based training interventions to reduce obstetric violence and, ultimately, improve patient care.

## 1. Introduction

Abusive and disrespectful treatment of patients in health care has been increasingly documented over the past decades. Qualitative and quantitative studies indicate the wide reach of the phenomenon and the long-lasting suffering it can create for patients [1,2,3,4]. The concept of obstetric violence, a particular kind of abuse in health care (AHC), has focused attention specifically on the disrespect, abuse and mistreatment that women experience in pregnancy, childbirth and the immediate post-partum period [3,5]. Obstetric violence captures medical interventions, such as episiotomies and caesarean sections, performed without evidence of medical necessity or appropriateness and/or patient consent [6]. It includes women’s perceptions of dehumanisation, discrimination, humiliation and other types of mistreatment related to health provider actions such as shouting, deliberate ignoring, denying pain relief, slapping, gossiping. Additionally, the concept of obstetric violence focuses attention on broader health system conditions that contribute to poor treatment or quality of care, such as gender inequities and other hierarchies of power, degrading working conditions, and insufficient policy frameworks to ensure patient rights [6,7,8].

The use of abusive behaviours against women during childbirth in delivery facilities has been the subject of numerous studies, predominately in developing countries [9,10,11,12,13,14,15]. It has been shown to have negative impacts on the overall quality of care and trust in health providers and systems, and is associated with adverse maternal and neonatal outcomes [16]. Obstetric violence is a breach of the fundamental task of health care to alleviate suffering [17,18]. The World Health Organisation (WHO) has condemned abuse, neglect or disrespect of patients during childbirth as a violation of internationally adopted human rights standards and principles [19]. Women’s right to the highest attainable standard of health includes the rights to be free from violence and discrimination and to dignified, respectful health care throughout pregnancy and childbirth [20].

This paper responds to a global call to action for respectful maternity care [20]. In prior research in Sri Lanka, we documented the existence of obstetric violence in public health facilities and a corresponding need to sensitise health providers about the phenomenon and its repercussions [21]. In the current paper, we report the findings from a pilot intervention study in Sri Lanka’s capital district, Colombo. The aim of the study was to assess the possibilities for use of a participatory theatre technique called Forum Play (FP) in training workshops as a method to reduce and prevent AHC in maternity care contexts. 

## 2. Materials and Methods

We invited physicians and nurses working in obstetrics and gynaecology units at three public hospitals in Colombo district to participate in half-day FP training workshops. One of our Sri Lankan co-authors distributed written information about the study on the hospital wards in advance of the workshops. The hospital directors followed up with verbal information about the study, gave formal permission and arranged transportation for all off-duty staff to attend the workshops on the days. 

We had previously developed the workshops for Swedish health care settings [22,23]. The pedagogy underlying the training draws from the works of Brazilian stage director, Augusto Boal, originator of the Theater of the Oppressed and Forum Theatre, and Paulo Freire on problem-posing dialogue and empowerment [24,25,26,27,28,29]. The methods developed by Boal and Freire have been applied over several decades now to curricula and training development to promote participation, personal and collective reflection, and transformative actions in oppressive or ethically-complex situations [30,31]. This includes applications in medical contexts, such as in medical and nursing education [32,33,34,35,36,37].

An experienced drama pedagogue led the FP workshops, which we held over the course of four consecutive days in November 2017. The workshops began with warm-up exercises to build comfort and rapport amongst and between the participants and research team. Thereafter, the core activities started with sharing personal experiences of situations of AHC. These were collected on paper (Figure 1). The group then selected several of the situations to explore in the ‘plays’ by voting with pen marks on the papers (as seen in Figure 1). In each workshop, we re-enacted one or two of the chosen situations to demonstrate the problems visually and collaboratively and repeatedly re-play them to test potential alternative courses of action. Figure 2 is an example of one of the situations of AHC identified by the participants based on personal experiences that was explored in a workshop.

The first two workshops were conducted in English, with a total of 20 participating physicians. The remaining two workshops were conducted in English and Sinhala, with 30 nurses in total. One of our Sri Lankan authors provided language interpretation. We did not mix physicians and nurses in the workshops in an effort to avoid potential dominance and subordination related to the rigid professional hierarchies in Sri Lankan working environments. Each workshop was held in the same location, a research institute unrelated to the participants’ workplaces. 

Before and three-to-four months after the workshops, the participants completed a questionnaire developed by the research team. This included background information (for example, age, sex, professional category) and four questions assessing their views on AHC (Figure 3). Validated research instruments about AHC are scarce, and none exist for evaluating the kind of intervention we assessed in this study. We selected the four-items for the questionnaire from a longer assessment tool used in one of our Swedish studies [38]. These questions performed well in the Swedish study and were tested in a pilot study for suitability in Sri Lanka before baseline data collection. To ensure confidentiality, the survey responses were not associated with participant names nor identification numbers. We conducted chi-square and Fisher’s exact tests using the SPSS Statistical Package for the Social Sciences to obtain before and after comparisons.

All participants gave their informed written consent for inclusion in the study before the workshops and were told that their participation could be withdrawn at any time without explanation or consequence. The study was conducted in accordance with the Declaration of Helsinki, and the protocol was approved by the Ethics Review Committee (ERC), Faculty of Medical Sciences, University of Sri Jayewardenepura (project ID: 55/17).

## 3. Results

The workshop participants were mainly female, between the ages of 35–50, Singhalese, and had more than three years of work experience. We found no statistically significant differences in the background characteristics of the participants at follow-up, except regarding occupation; only five physicians and 25 nurses answered the questionnaire a second time (Table 1). A total of 80 questionnaires (50 at baseline and 30 at follow-up) were used for the comparative before (baseline) and after (follow-up) evaluation of the intervention.

In all, only two participants, both at baseline, reported that AHC had no relevance to them. At baseline, 78% of participants reported that they had only picked up some general aspects about AHC or heard about cases of AHC at their workplace, but were not personally involved in the situations. Our main statistical finding is that the baseline group, compared with themselves at follow-up, more often reported that they had been involved personally in cases of AHC (*p* = 0.035) (Table 2).

We found no statistically significant results for the remaining three questions. For the question about regret, we asked participants who reported that they had not been in a situation of AHC to continue to the next question. Due to these skip instructions, there were too few observations to draw conclusions, but the skip pattern dropped from 56% before to 33.3% after the workshop. Some other trends are notable. Namely, when asked about earlier responses in AHC situations, 44% reported that they had not been in such a situation before the intervention compared to 30% at follow-up, and 43% reported acting in support of the patients’ position compared to 35% before the workshop (Table 2).

## 4. Discussion

It is no simple task to challenge the entrenched and often mechanical or unconscious norms and patterns of behaviour that perpetuate AHC, particularly in under-resourced health systems where the availability of options to act differently may also be limited (for example, obstetric contexts lacking essential supplies, including the health workforce). Creative efforts have to be built from the ground up to address these challenges [39]. We brought FP to Sri Lanka at the invitation of one of our Sri Lankan co-authors, a public health doctor. We had previously used FP workshops with promising results in Sweden, both in terms of strengthening participants’ abilities to recognise AHC and improving their readiness to act against AHC in real-world situations [22,23,38,40]. The results of the current study indicate modest but promising evidence of the same benefits of the method in Sri Lanka, and also raise a number of questions to address in future research. 

First, the workshops reconfirmed that FP can be an effective strategy for increasing the capacity of health care providers to recognise obstetric violence. All participants recalled situations when a patient had been abused by a health care provider. Moreover, as each workshop progressed, the participants shared more personal stories of enacting and witnessing AHC. A recurring topic in the situations they described was the normalised way that staff respond to ‘uncooperative’ patients in childbirth (women described as not obeying staff or expressing pain, fear or desperation loudly) by scolding them, yelling at them, insulting them, and using physical force to punish, restrain and/or examine them. Women who were pregnant at a young age or outside of marriage, even if the result of rape, and women with sexually transmitted infections, were also identified as being particularly targeted with staff contempt, expressed openly through verbal abuse.

Following the FP intervention, participants more often reported that they had been personally involved in situations of AHC compared to before the workshop. We also witnessed promising trends in the data in terms of increased recognition of the importance of considering patient perspectives in medical care. This indicates an increase in participants’ awareness of AHC, or in their abilities to conceptualise the phenomenon of AHC, or possibly both. To clarify this distinction, participants may have lacked the professional vocabulary and practice for identifying and conceptualising AHC before the workshop. Alternatively, they may not have identified with the term because of an assumption that it implied the actions were intentional. Participants may have also had some awareness of the range of dilemmas that AHC created in their daily work but may not have reflected on its frequency, impact on patients, or alternative ways of acting or behaving in these situations.

Recognising AHC is an important step towards improving readiness to take action against it in real-world situations. Our results also suggest that the FP method, which combines collaborative social interaction with practical or ‘hands-on’ learning, can encourage new ways of thinking and behaving in health care practice. In the workshops, the scenes that were played out were altered by the audiences’ interventions; that is, the workshop participants rehearsed realistic solutions and new paths to responding to the scenes of AHC. In these rehearsals, instead of being tolerated, accepted or even reinforced, the abuse was rejected and the patient’s dignity protected. Notably, protecting dignity was also considered necessary to restore confidence in health care. Thus, the FP method reinforced that it is not only the situation of abuse, but also its prevention, that can be undertaken collaboratively.

The results from this pilot study are persuasive regarding the feasibility and value of the training method. The workshop participants expressed their appreciation for the training and requested more training, for themselves and their colleagues who were not present, as well as staff at other hospitals. However, the conclusions that can be drawn from this study are limited by a variety of factors. Namely, the study is small, with only a few participants, and only one workshop for each participant. The involvement of a control group would have allowed us to assess the amount of change before and after the intervention for each group separately, in turn strengthening our results. Also, a variety of types of interventions are needed to tackle the complex structural drivers of AHC that cannot be addressed at the individual level, nor even the level of singular hospitals. AHC can result from the individual behaviours and attitudes of health providers, but our studies in Sweden and Sri Lanka suggest that contextual factors are more critical for the occurrence of AHC than are individuals and their actions or non-actions [21,41]. This perspective aligns with increasing empirical evidence from other study sites around the world, which also demonstrate the embeddedness of AHC in complex intersections of social and cultural dynamics, political histories, and family, community and institutional power structures [39,42].

This study also identified numerous questions to address and approaches to refine in related future training interventions, which may be of relevance in other low-resource contexts and/or regional settings. For example, is a training intervention a potentially effective strategy to change the norms and standards of care that allow for the existence of such types of AHC, such as attitudes that discriminate and devalue women in society at large? Can we ethically justify workshops on counteracting AHC if staff risk being abused for stepping up to counter AHC when they witness it, or in environments lacking policies and structures to effectively ensure both patient and employee rights? What are the long-term effects of such an intervention? How can we ensure that changes in attitudes and behaviours from such an intervention are enduring and sustainable? Will training health providers lead to improved maternal and neonatal health outcomes and patient assessments of quality of care? More experimental evidence is required to answer these and related questions. 

## 5. Conclusions

A brief training intervention using improvisational and participatory theatre techniques may be an efficient ‘eye-opener’ and safe forum for discussing the individual behaviours, cultures and structures in the health system that allow for disrespect and abuse of patients. Our results also suggest that FP has the potential to encourage new ways of thinking to prevent and reduce AHC. This observation indicates possibilities for further developing strategies to counteract AHC, even in very hierarchical health care systems, as in Sri Lanka. 

## Figures and Tables

**Figure 1 ijerph-16-01616-f001:**
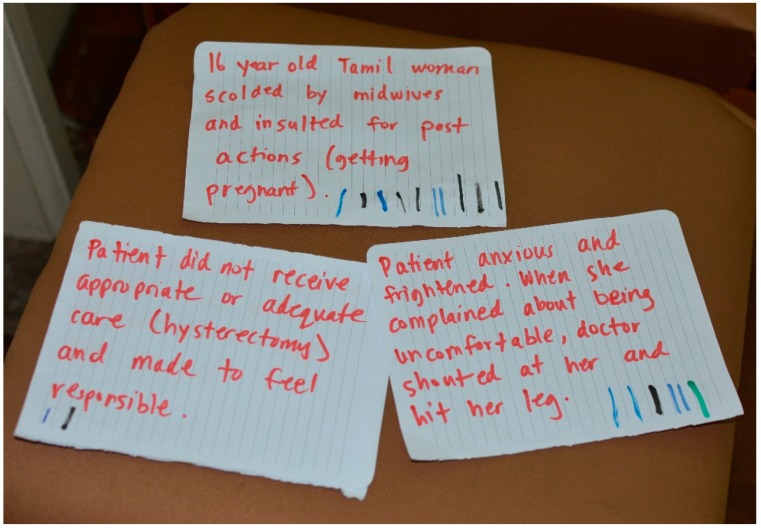
Examples of scenarios of abuse in health care identified by workshop participants.

**Figure 2 ijerph-16-01616-f002:**
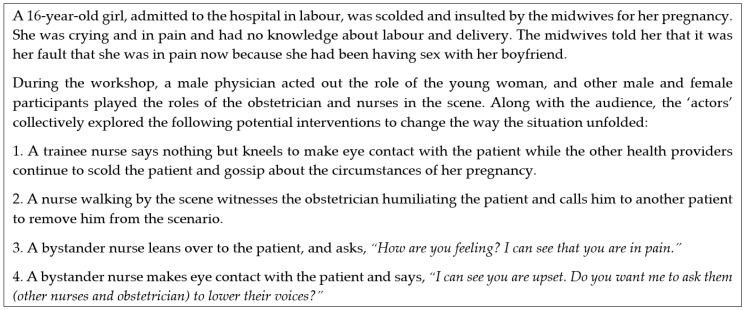
Example of a typical situation described by participants as illustrative of abuse in health care, and suggested solutions explored during the intervention.

**Figure 3 ijerph-16-01616-f003:**
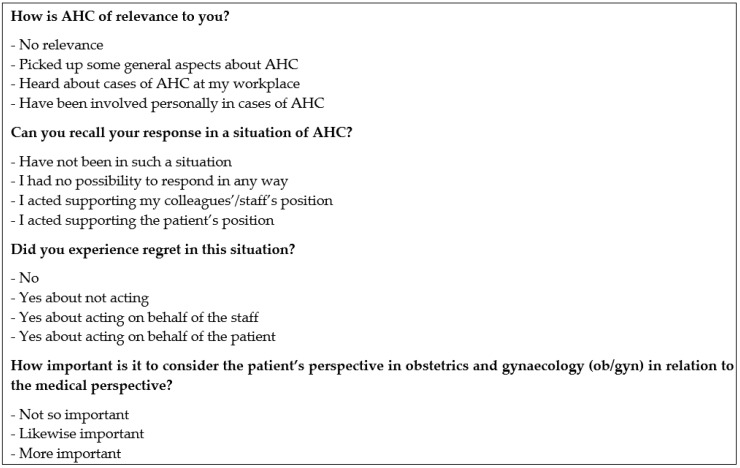
Questionnaire developed to assess participants’ views of abuse in health care and used for the intervention evaluation.

**Table 1 ijerph-16-01616-t001:** Background characteristics of study participants at baseline and follow-up time points (*n* = 80).

Variable	Response Options	Baseline	Follow-up	*p*-Value
		*n* = 50	*n* = 30	
		*n* (%)	*n* (%)	
Sex	Female	41(82.0)	27(90.0)	
	Male	9(18.0)	3(10.0)	
				0.520
Age	19–34	14(28.0)	2(6.7)	
	35–50	30(60.0)	21(70.0)	
	51–62	6(12.0)	7(23.3)	
				0.049
Ethnicity	Singhalese	47(94.0)	29(96.7)	
	Tamil	1(2.0)	0(0.0)	
	Moor or Muslim	2(4.0)	1(3.3)	
	Other	0(0.0)	0(0.0)	
				0.728
Occupation	Physician	20(40.0)	5(16.7)	
	Nurse	30(60.0)	25(83.3)	
				0.045
Work experience (years)	0–2	3(6.0)	0(0.0)	
	3–12	25(50.0)	10(34.5)	
	13–22	16(32.0)	13(44.8)	
	23–33	6(12.0)	6(20.7)	
	Missing		1	
				0.215

**Table 2 ijerph-16-01616-t002:** Questions on abuse in health care used for evaluation (*n* = 80).

Variable	Response Options	Baseline	Follow-up	*p*-Value
		*n* = 50	*n* = 30	
		*n* (%)	*n* (%)	
How is AHC of relevance to you?	No relevance	2(4.0)	0(0.0)	
	Picked up some general aspects about AHC	24(48.0)	7(23.3)	
	Heard about cases of AHC at my workplace	15(30.0)	10(33.3)	
	Have been involved personally in cases of AHC	9(18.0)	13(43.3)	
				0.035
Can you recall your response in a situation of AHC?	Have not been in such a situation	21(43.8)	9(30.0)	
	I had no possibility to respond in any way	8(16.7)	6(20.0)	
	I acted supporting my colleagues’/staff’s position	2(4.2)	1(3.3)	
	I acted supporting the patient’s position	17(35.4)	13(43.3)	
	Missing	2	1	
				0.565
Did you experience regret in this situation?	No	12(40.0)	13(43.3)	
	Yes about not acting	8(26.7)	5(16.7)	
	Yes about acting on behalf of the staff	1(3.3)	1(3.3)	
	Yes about acting on behalf of the patient	1(3.3)	1(3.3)	
	Missing ^1^	28(56.0)	10(33.5)	
				0.964
How important is it to consider the patient’s perspective in obstetrics and gynaecology (ob/gyn) in relation to the medical perspective?	Not so important	0(0.0)	0(0.0)	
	Likewise important	17(37.8)	16(53.3)	
	More important	28(62.2)	13(43.3)	
	Missing	5	1	
				0.160

^1^ Dropout due to skip instruction: ‘Have not been in such a situation?’ (30/80).

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
