# Peer review of "Can Forum Play Contribute to Counteracting Abuse in Health Care? A Pilot Intervention Study in Sri Lanka"

_ijerph, 2019, doi:10.3390/ijerph16091616_

Round 1

Reviewer 1 Report

Dear Authors,

I enjoyed reading your paper and I believe that the paper discusses an important issue in healthcare particularly in developing countries. While the focus of the paper is noble, and it is a well-written paper, there are certain aspects of the paper, particularly the study methodology, that raise concerns about the study findings and the claims made based on those studies. I do not believe the paper warrants publication in this format, however, if you wish to improve your paper, I provided several suggestions to consider for the paper’s revision.

Introduction Section:

The background section of the paper is well-written and provide sufficient information for the reader to understand the issue.

Methods Section:

First, the recruitment procedure of the participants was not described, because how participants were recruited will determine if participants differ from others who did not participate in the study. Also, the workshop is described, no info is registered as to what data was collected during the workshop. In the discussion section, the authors referred to the participant's responses and stories related to the topic, but no info was provided in this section. If the authors wish to report participant stories/conversations and observations by the researchers as qualitative data, they should be described in this section. Why the notes by participants as displayed on the photo on page 3 were not used as qualitative data?

The major concern related to the methodology is the pre-and post-questionnaire that included only four items. There is no justification regarding the limited items in the assessment tool, and how they were selected, and why this scaling method for the items was chosen. The choice of the items in the questionnaire is not clear.

On page 4, the authors stated 80 questionnaires were used, the sample size of the study was 50, so not clear where this number is coming from? Also, the participant descriptions should be moved from the methods to the beginning of the results section.

In table 1, one of the groups for the ethnicity is listed as “Muslim”, how does religious affiliation qualify to be an ethnicity?

Results Section:

The second sentence in this section, the results should be reported in percentages rather than numbers.

Conclusion Section:

Page 7, lines 223-224, the authors state that “ we observed an increase in participants’ awareness of AHC or in their abilities to conceptualize the phenomenon of AHC, or possibly both”. How do these claims supported by the study results? According to Table 2, these are not an accurate interpretation of the study findings. How was the conceptualization of the phenomenon of AHC assessed in this study to make this claim?

In addition to that, there is new info presented in this section, as well as claims, are made that are not supported by this study’s findings.  

I hope these comments will be helpful,

Best of luck!

Author Response

The attached pdf file contains our responses to all three reviewers. Below are our specific responses to Reviewer 1.

Methods Section:

First, the recruitment procedure of the participants was not described, because how participants were recruited will determine if participants differ from others who did not participate in the study.

Authors’ response: Our Sri Lankan host and co-author visited the three hospitals to meet the directors and discuss the study. She distributed written information sheets about the study on the prenatal, delivery and postal units/wards of the hospitals. After we confirmed the dates for the workshops, the hospital directors gave formal permission to the off-duty nurses and physicians to attend on those days (i.e. the staff who were not on roster or shift). The directors also arranged transportation from the hospitals to the location where we held the workshops to facilitate attendance. We cannot speak to the representativeness of the sample of individuals who ultimately participated in the workshops, but they varied in age, gender, years of work experience and we have no reason to believe they differed in any particular ways from the staff who were working/on-duty on the workshop days. We have added some additional information into the manuscript now, as follows (lines 78-81 track changes version):

“One of our Sri Lankan co-authors distributed written information about the study on the hospital wards in advance of the workshops. The hospital directors followed up with verbal information about the study, gave formal permission and arranged transportation for all off-duty staff to attend the workshops on the days.”

The workshop is described, but no info is registered as to what data was collected during the workshop. In the discussion section, the authors referred to the participant's responses and stories related to the topic, but no info was provided in this section. If the authors wish to report participant stories/conversations and observations by the researchers as qualitative data, they should be described in this section. Why the notes by participants as displayed on the photo on page 3 were not used as qualitative data?

Authors’ response: The discussion section has been re-worded to remove some of the ambiguity mentioned here (e.g. reference to participant responses). The scenarios of AHC and notes displayed in the photo are not part of our data but rather a vital part of the procedure of the workshop. Forum Play starts from collective sharing of personal experiences, which become the content for collaborative reflection and problem-solving through rehearsing potential alternative courses of action. We feel it is valuable to include examples of the notes and scenarios (Figures 1 and 2) to illustrate the tools and methods of the workshops. Thank you for bringing this to our attention; it was not clearly described in our original materials and methods section. We have changed the order of the figures in the revised manuscript now (swapped the two figures), and explained each of the figures in more detail. The revised text reads as follows (lines 91-99 track changes version):

“The workshops began with warm-up exercises to build comfort and rapport amongst and between the participants and research team. Thereafter, the core activities started with sharing personal experiences of situations of AHC. These were collected on paper (Figure 1). The group then selected several of the situations to explore in the ‘plays’ by voting with pen marks on the papers (as seen in Figure 1). In each workshop, we re-enacted one or two of the chosen situations to demonstrate the problems visually, and collaboratively and repeatedly re-play them to test potential alternative courses of action. Figure 2 is an example of one of the situations of AHC identified by the participants based on personal experiences that was explored in a workshop.”

The major concern related to the methodology is the pre-and post-questionnaire that included only four items. There is no justification regarding the limited items in the assessment tool, and how they were selected, and why this scaling method for the items was chosen. The choice of the items in the questionnaire is not clear.

Authors’ response: There are no established instruments for evaluating this kind of intervention. Therefore, we chose a selection of questions that were used, and performed well, in a previous Swedish study. In the Swedish study, we started out with many questions (three pages) and ended up with a single page, making modifications over the course of the data collection period. The reason for shortening the questionnaire was to minimise drop-out given that the intervention study lasted over one year and had four follow-up time points (with losses each time). Here, in Sri Lanka, we had only one opportunity for data collection, so we decided to use the shortest (last) version of our prior questionnaire. Our Sri Lankan collaborators carried out a small pilot test of the four-item questionnaire prior to our baseline data collection to ensure its appropriateness and comprehension. We have added this to the materials and methods section as follows (lines 135-141 track changes version):

“Before and three-to-four months after the workshops, the participants completed a questionnaire developed by the research team. This included background information (for example, age, sex, professional category) and four questions assessing their views on AHC (Figure 3). Validated research instruments about AHC are scarce, and none exist for evaluating the kind of intervention we assessed in this study. We selected the four-items for the questionnaire from a longer assessment tool used in one of our Swedish studies [38]. These questions performed well in the Swedish study and were tested in a pilot study for suitability in Sri Lanka before baseline data collection.”

On page 4, the authors stated 80 questionnaires were used, the sample size of the study was 50, so not clear where this number is coming from?

Authors’ response: There were 50 questionnaires at baseline and 30 at follow-up, thus 80 in total, as indicated in Tables 1 and 2. This is clarified also in the text of the manuscript now, see line 195 in track changes version of the document.

Also, the participant descriptions should be moved from the methods to the beginning of the results section.

Authors’ response: We have made this change now. Thank you for the constructive input.

In table 1, one of the groups for the ethnicity is listed as “Muslim”, how does religious affiliation qualify to be an ethnicity?

Authors’ response: In Sri Lanka, the term “Muslim” is used colloquially, as well as in many official documents, to describe an ethnic group that comprises all individuals who practice the religion of Islam (e.g. Indian Moors, Sri Lankan Moors, Malays). The terms “Moor” and “Muslim” are often used interchangeably to describe ethnicity. After consultation with our Sri Lankan co-authors, we have revised Table 1 now to include Moor, thus the ethnicities in the Table are now written as: Singhalese, Tamil, Moor or Muslim, Other.

Results Section:

The second sentence in this section, the results should be reported in percentages rather than numbers.

Authors’ response: Yes, agree. We have changed this (now line 203 in track changes version).

Conclusion Section:

Page 7, lines 223-224, the authors state that “we observed an increase in participants’ awareness of AHC or in their abilities to conceptualize the phenomenon of AHC, or possibly both”. How do these claims supported by the study results? According to Table 2, these are not an accurate interpretation of the study findings. How was the conceptualization of the phenomenon of AHC assessed in this study to make this claim? In addition to that, there is new info presented in this section, as well as claims, are made that are not supported by this study’s findings.  

Authors’ response: At follow-up, participants more often reported they had been involved in situations of AHC than at baseline. We interpret this as an indication of new awareness of the phenomenon of AHC or new language/other ability to conceptualise AHC. We have modified the order of the content in the Discussion now, so the section now begins with summary interpretations of our key results and then discusses the limitations and questions for future research. The Conclusion is also now significantly shortened, following advice from you and Reviewer 3. We are confident these revisions have increased the clarity of our claims about participants’ awareness of AHC, and have addressed your comment that the original conclusion introduced new and unsupported claims.

Reviewer 2 Report

A very interesting read. The paper reports on the pilot testing of a novel intervention (Forum Play), which uses participatory theatre techniques to raise awareness and help to improve the ability of healthcare workers in Sri Lanka to tackle obstetric violence in public hospitals.

I thought this was a fascinating approach and a very well-written and clear paper overall.

I have a few comments/suggestions:

1.       To prevent confusion, you may wish to add the word ‘medical’ before ‘interventions’ on Line 48;

2.       The sentences in Lines 71-75 feel a bit out of place in the Introduction. I would recommend to stop at the aim and move these few sentences to the first paragraph of the Discussion, which in my opinion, could benefit from a few summary sentences of what the study was about and what was found.

3.       Why were the workshops not mixed, including both doctors and nurses together? I suspect I know the answer this question, but it would be helpful to be explicit about the reasoning in the text.

4.       It would be really interesting as well (and maybe this is something to consider in your future work) why so many doctors were lost to follow-up. Did the authors do anything specific to engage doctors? And keep their interest sustained? I would be keen to know more about acceptability of the intervention, and whether it was anything about the approach that prevented them from continuing their participation by completing all of the questionnaires. Or whether it was just time commitment required for the study that prevented them from staying involved.

5.       Noticed a missed word on Line 217 – ‘be built.’

Author Response

The attached pdf file contains our responses to all three reviewers. Below are our specific responses to Reviewer 2.

To prevent confusion, you may wish to add the word ‘medical’ before ‘interventions’ on Line 48. Authors’ response: Yes, done. Thank you.

The sentences in Lines 71-75 feel a bit out of place in the Introduction. I would recommend to stop at the aim and move these few sentences to the first paragraph of the Discussion, which in my opinion, could benefit from a few summary sentences of what the study was about and what was found. Authors’ response: Yes, we agree too. We originally wrote these sentences because the journal guidelines for the Introduction state to “identify the main aim of the work and highlight the main conclusions.” However, conclusions seem out of place in the Introduction, so these have been integrated into the revised Discussion section now. We have also started the Discussion with a summarising paragraph, as you suggested. Thank you for the helpful input.

Why were the workshops not mixed, including both doctors and nurses together? I suspect I know the answer this question, but it would be helpful to be explicit about the reasoning in the text. Authors’ response: This was suggested by our Sri Lanka hosts and co-authors, given the strict nature of professional roles and hierarchies in the country. The workshop itself, with mixed genders, was already considered experimental, thus we kept the professional groups separate to reduce some potential dominance/subordination related to professional status and hierarchy. A line to this effect has been added in the materials and methods section now (lines 131-133 track changes version): “We did not mix physicians and nurses in the workshops in an effort to avoid potential dominance and subordination related to the rigid professional hierarchies in Sri Lankan working environments.”

It would be really interesting as well (and maybe this is something to consider in your future work) why so many doctors were lost to follow-up. Did the authors do anything specific to engage doctors? And keep their interest sustained? I would be keen to know more about acceptability of the intervention, and whether it was anything about the approach that prevented them from continuing their participation by completing all of the questionnaires. Or whether it was just time commitment required for the study that prevented them from staying involved. Authors’ response: Engagement from the doctors was high during the workshop. We also held focus group discussions with a selection of the participants following the workshops. These are currently being analysed for a future qualitative paper from this pilot intervention. All indications from these discussions are of high acceptability of the intervention from the doctors. The higher loss to follow-up from doctors than nurses is most likely due to practical/logistic reasons which we will need to address in future studies; primarily, the time between the workshop and follow-up was likely too long. Nurses rarely change place of work in Sri Lanka unless they change household residence, but the mobility of doctors is high. Many doctors are attached to hospitals on temporary assignments during postgraduate studies and others act as consultants in different institutions on a regular basis. It is also possible that the doctors’ enthusiasm for the study was lower compared to nurses. Nurses are more powerless to effect change or intervene when they witness AHC in the rigid professional hierarchy in Sri Lanka. We can speculate that this may make the topic more relevant and important for nurses than for doctors. 

Noticed a missed word on Line 217 – ‘be built.’ Authors’ response: Fixed. Thank you!

Reviewer 3 Report

A brilliant article adding a credible contribution to the much desired contemporary quest of addressing respectful maternity care.

The only one thing I would request authors to consider changing is the Conclusion. It is unduly long for the length of the article and a bit unfocused. It contains more description of results, study limitations and suggestions for future research. 

I would recommend a succinct summary of the key message(s) from the study. A suggestion given below (taken from line 231 -245): 

Our results indicate that FP is an efficient ‘eye-opener’ and a safe forum for discussing the individual behaviours, cultures and structures in the health system that allow for disrespect and abuse of patients. Our findings also suggest that FP has the potential to encourage new ways of thinking to prevent and reduce AHC. This observation indicates possibilities for further developing strategies to counteract AHC, even in very hierarchical health care systems, as in Sri Lanka.

The rest of the information in the conclusion especially the questions raised, is also useful and could be moved up in the discussion section or - journal guidelines permitting - placed in two separate sections after discussion; study limitations and further research.

Author Response

The attached pdf file contains our responses to all three reviewers. Below are our specific responses to Reviewer 3.

A brilliant article adding a credible contribution to the much desired contemporary quest of addressing respectful maternity care. The only one thing I would request authors to consider changing is the Conclusion. It is unduly long for the length of the article and a bit unfocused. It contains more description of results, study limitations and suggestions for future research. I would recommend a succinct summary of the key message(s) from the study. A suggestion given below (taken from line 231 -245):

Our results indicate that FP is an efficient ‘eye-opener’ and a safe forum for discussing the individual behaviours, cultures and structures in the health system that allow for disrespect and abuse of patients. Our findings also suggest that FP has the potential to encourage new ways of thinking to prevent and reduce AHC. This observation indicates possibilities for further developing strategies to counteract AHC, even in very hierarchical health care systems, as in Sri Lanka.

The rest of the information in the conclusion especially the questions raised, is also useful and could be moved up in the discussion section or - journal guidelines permitting - placed in two separate sections after discussion; study limitations and further research.

Authors’ response: Perfect! Thank you very much for the positive review and constructive input. We have shortened the Conclusion now, and restructured the content in the Discussion to improve the clarity and flow. The journal guidelines recommend no sub-headings so we have grouped study limitations and questions for further research into two discrete paragraphs, the final two at the end of the Discussion section.